# A Performance Assessment Study of Different Clinical Samples for Rapid COVID-19 Antigen Diagnosis Tests

**DOI:** 10.3390/diagnostics12040847

**Published:** 2022-03-29

**Authors:** Naveed Ahmed, Mohammad Nur Amin Kalil, Wardah Yusof, Mimi Azliha Abu Bakar, Afifah Sjamun Sjahid, Rosline Hassan, Mohd Hashairi Fauzi, Chan Yean Yean

**Affiliations:** 1Department of Medical Microbiology and Parasitology, School of Medical Sciences, Universiti Sains Malaysia, Kubang Kerian 16150, Malaysia; namalik288@gmail.com (N.A.); wardahyusof@usm.my (W.Y.); 2Department of Emergency Medicine, School of Medical Sciences, Universiti Sains Malaysia, Kubang Kerian 16150, Malaysia; aminkalil28@student.usm.my (M.N.A.K.); azliha79@usm.my (M.A.A.B.); afifahkk@usm.my (A.S.S.); 3Hospital Universiti Sains Malaysia, Universiti Sains Malaysia, Kubang Kerian 16150, Malaysia; roslin@usm.my; 4Department of Hematology, School of Medical Sciences, Universiti Sains Malaysia, Kubang Kerian 16150, Malaysia

**Keywords:** SARS-CoV-2, real-time RT-PCR, early diagnosis, diagnostic kit, performance accuracy, rapid self-test

## Abstract

Accurate diagnosis to limit the spread of SARS-CoV-2 is crucial for the clinical management of this lethal infection. Recently, many low-cost and easy-to-use rapid test kits (RTK) have been developed in many countries for the massive screening of SARS-CoV-2. Thus, evaluating the accuracy and reliability of an RTK is critical. The current study was conducted on 157 individuals to evaluate the performance accuracy of rapid SARS-CoV-2 antigen detection kits using different clinical samples compared with qRT-PCR results. Nasopharyngeal swabs were collected from patients for qRT-PCR and RTK tests, and then buccal and nasal, and nasal swabs were collected for RTK tests separately. The nasal and buccal swabs showed high sensitivity (98%) and specificity (100%) compared with the qRT-PCR results. Meanwhile, for nasal, the sensitivity was 96% with 98% specificity, and nasopharyngeal swabs showed 98% sensitivity and 94% specificity. Fisher’s exact test revealed statistical significance (*p* < 0.05) between nasopharyngeal, nasal and buccal, and nasal swabs compared with qRT-PCR results. The study concludes that different clinical samples used for the rapid diagnosis of SARS-CoV-2 showed high sensitivities and specificities compared with qRT-PCR. The RTKs using nasal and buccal, nasopharyngeal, and nasal swabs are valuable tools for the early detection of SARS-CoV-2, especially when molecular detections are available with limited access and a high infectivity rate, when the timely detection of virus cases is urgently needed. These types of clinical samples are effective to be used by RTKs for surveillance among community and healthcare workers.

## 1. Introduction

The Severe Acute Respiratory Syndrome Coronavirus-2 (SARS-CoV-2) pandemic has created significant public health problems globally since the first confirmed cases were reported in December 2019 [1,2]. To help minimize this pandemic, the early identification and isolation of patients and their contacts are critical. The standard diagnostic approach for SARS-CoV-2 infection, also known as coronavirus disease 2019 (COVID-19), is real-time reverse-transcription polymerase chain reaction (qRT-PCR), which was first introduced in January 2020 for COVID-19 diagnosis [3,4]. This approach is based on reliable, quick, and simple diagnostic instruments that can evaluate large numbers of samples in a short amount of time [3].

Depending on the type of methods or kits used, the qRT-PCR takes 1–4 h to complete. In addition, it must be carried out by trained molecular personnel [4]. A significant constraint of existing public health containment methods is the huge gap between the high number of patients/contacts and laboratory capabilities to conduct qRT-PCR in a timely way [5]. Furthermore, several regions have experienced qRT-PCR reagent shortages. As a result, other assays are in high demand, such as antigen detection tests, which, unlike antibody tests, may identify the presence of the virus itself in buccal or nasal swabs [6].

To help contain this pandemic, early identification of cases is critical; therefore, fast and simple-to-use diagnostic technologies that can analyze large quantities of samples in a short amount of time are necessary [7,8]. Rapid and accurate SARS-CoV-2 assays are required to accelerate disease prevention and control, as well as screening during pre-operative care for invasive operations [9].

Antigen detection tests offer the benefit of identifying the virus itself and, therefore, may be a superior tool for early cases, but they need a high-quality sample and adequate viral loads [10]. If the accuracy of lateral flow immunoassays employing monoclonal anti-SARS-CoV-2 antibodies, which target SARS-CoV-2 antigens, is similar to that of real-time RT-PCR assays, they may be used as a supplement to qRT-PCR assays [11]. Even though the first commercial assay for SARS-CoV-2-specific antigen testing received regulatory approval in mid-2020 [12], the market pressure created by this unprecedented pandemic has resulted in many new assays being commercially available. Unfortunately, there is a lack of scientific literature supporting their accuracy and real-time performance using different types of clinical samples. Thus, validation and comparison are needed [13,14].

Other methods include serological testing, which is presently not indicated for case identification owing to the diagnostic limits in early infections. Rapid diagnostic tests (RDTs) should be prioritized among test formats since they are speedy, simple to conduct, and may be used as point-of-care testing [15,16].

To prevent the spread of SARS-CoV-2 infection in the community, early identification, efficient isolation of symptomatic patients, and thorough tracking of close contacts are critical [17,18]. Antigen rapid test kits (RTKs) are ideal for point-of-care testing (POCT) because they can be carried out and interpreted without any special equipment, are cheap, and reduce turnaround times [16,19]. Currently, the large gap between the high number of cases and the qRT-PCR testing capacity is a major limitation, resulting in a need for alternative assays such as RTKs, which may identify the presence of the antigen in different respiratory samples. Many types of RTKs have been recently developed and made commercially available. However, the information about the performance efficacy of these assays with different clinical samples is limited. Keeping in mind the feasibility of RTKs with different clinical samples, we conducted the current study to evaluate the accuracy of different respiratory samples used for COVID-19 early diagnosis by various RTKs. The efficacy of different clinical samples (with RTK test) was compared with the qRT-PCR.

## 2. Materials and Methods

### 2.1. Study Design and Setting

A cross-sectional evaluation single-center study was conducted on 157 suspected cases of COVID-19 infection, who were either close contacts of confirmed COVID-19 infected patients or showed COVID-19-like signs and symptoms. The suspected cases were recruited from Hospital Universiti Sains Malaysia (HUSM), Kelantan, Malaysia, from 13 August 2021 to 31 December 2021. There were no exclusion criteria for any age group, gender, race, ethnicity, other co-morbidities, etc. The study participants’ demographic characteristics (sex, age, and ethnicity) were collected on a written informed consent form. Only individuals who agreed to sign the written informed consent document proceeded to the current study. This study was conducted under the ethical approval of the Human Research Ethics Committee of Universiti Sains Malaysia (USM) (JEPeM) (Ethical Approval No: USM/JEPeM/COVID19–44, approved on 19 July 2020).

### 2.2. Clinical Sample Collection

A total of 157 patients were recruited for the current study to evaluate the specificity, sensitivity, and accuracy of RTK tests using different types of clinical samples. The patient diagnosis data were counter-checked with Malaysia’s Public Health Laboratory Information System database (SIMKA) (https://simka.moh.gov.my/, accessed on 15 November 2021) (Ministry of Health, Malaysia). The sample collection and disposal of the kit were conducted according to the standard operating procedures (SOPs) for COVID-19 as recommended by the World Health Organization (WHO). Table 1 shows the number of samples processed for RTKs and qRT PCR.

Nasopharyngeal (NP) swabs (collected by healthcare professionals) were collected in viral transport medium (VTM) (Global Science Sdn Bhd, Jalan, Malaysia) from the individuals, and then the nasal and buccal and nasal swabs (collected by patients themselves/self-collect) were collected separately. The collected samples were immediately transported to the COVID-19 diagnostic laboratory for further processing. After NP swabs were collected by healthcare professionals, each sample were proceeded to RTK (Panbio, Abbott, IL, USA) and qRT-PCR. However, for RTK self-test kit (ProDetect™, Medical Innovation Ventures Sdn Bhd, Penang, Malaysia), the nasal swabs, a sterile swab was put into the left and right nostrils one at a time until minimal resistance was felt (approximately 2 cm inside of the nose). Against the nasal wall, the swab was spun 5–10 times. The swab was carefully withdrawn from the nasal canal and mixed with the buffer when the operation was completed. A fresh sterile swab was used to obtain buccal swab samples. Swab was inserted into the oral cavity between the cheeks (left and right) and the gums by rubbing up and down at least 4–6 times, and mixed with the buffer. The subjects were instructed not to put anything into their mouths for at least 10 min before the buccal swab collection, including drinks, tobacco products, or food. After collection of nasal and buccal and nasal swabs, the samples proceeded RTK self-test kit (ProDetect™, Medical Innovation Ventures Sdn Bhd, Malaysia).

### 2.3. RTK Method

The COVID-19 Antigen RTK is a membrane-based immunoassay for detecting SARS-CoV-2 nucleocapsid protein antigens in human respiratory samples (nasal, buccal and nasal, and NP swabs). The specimen interacts with SARS-CoV-2 nucleocapsid protein antibody-coated particles in the test device’s line area during the test. The combination migrates upwards on the membrane chromatographically and interacts with the antibody in the test line area by capillary action. A colored line appears in the test line region if the specimen contains SARS-CoV-2 antigens [20].

### 2.4. Testing Procedure

The test was performed immediately after sample collection. The test device was placed on a flat and clean surface. Two drops of the extraction sample were placed on the test device and left for 15 min at room temperature. The results appeared as colored lines. The test device was discarded in a biosafety box. The test results were interpreted as positive, negative, or invalid, as shown in Figure 1.

### 2.5. Real-Time RT-PCR Detection of SARS-CoV-2 

The qRT-PCR was performed using three different assays: Lytestar 2019-nCoV RT-PCR Kit (ADT Biotech Sdn Bhd, Malaysia), Labsystems (Labsystem Diagnostic, Vantaa, Finland), and Cepheid Xpert Xpress SARS-CoV-2 assay on the GeneXpert system (Cepheid, Sunnyvale, CA, USA). The Lytestar kit targets the amplification of *Envelope* (*E*) and *RNA-dependent RNA polymerase* (*RdRP*) genes, Labsystems targets *E* and *Open Reading Frame 1ab* (*ORF1ab*), while the GeneXpert targets the *E* and Nucleocapsid 2 (*N2*) genes. All systems were run with an internal control to rule out false negatives for confirmation.

### 2.6. Statistical Analysis

The data were subjected to statistical analysis to check the significance, sensitivity, specificity, accuracy, positive predictive value (PPV), and negative predictive value (NPV) of the tested clinical samples. SPSS version 26.0 was used for data analysis. Chi-square test and Fisher’s exact test were used to determine the significance of the data. A *p*-value of <0.05 was considered statistically significant.

## 3. Results

In the current study, from 157 recruited patients, *n*-90 were male, and *n*-67 were female. The participants were *n* = 140, *n* = 13, and *n* = 4 from the Malay, Chinese, and Indian ethnicities, respectively. A total of 102 were diagnosed as positive for COVID-19 with RTK tests using different types of clinical samples. After the confirmation of samples for PCR results, a total of 102 patients were found positive for COVID-19 infection, but with a difference in six cases. The detailed demographic characteristics of the study participants are shown in Table 2.

Among the tested samples, the nasal and buccal samples showed high sensitivity (98%) and specificity (100%) in relation to qRT-PCR. The sensitivity, specificity, accuracy, PPV, NPV values, and percentages of each tested samples are shown in Table 3.

The diagnostic accuracy of all tested clinical samples was found statistically significant (*p* < 0.05) in relation to RT-PCR. From the total 102 RT-PCR positive cases, a total of 96 were also positive using nasal and buccal, nasal, and NP swabs. The detailed comparison between different respiratory samples and qRT-PCR is shown in Table 4.

The overall Ct values obtained from qRT-PCR are shown in Table 5, which also shows the comparison of Ct values with the positive ratio of different clinical samples. No statistical difference was found when comparing the results of tested clinical samples with Ct values (Table 5). The efficiency of these tested samples was found even in Ct values of 38.1 to 41.5. Furthermore, the highest efficiency of the test device was noted in Ct values of <20.00.

## 4. Discussion

The importance of antigen-based RTKs has been proven in many recent studies, including their ease of use and low cost [16,21,22,23,24]. These RTKs can be used for the extensive screening and epidemiological surveillance of COVID-19’s spread [22,25,26]. Using these RTKs, the tests can be run using different types of clinical samples. The current study was conducted to evaluate the performance accuracy of RTKs using different types of clinical samples compared with qRT-PCR results. The RTKs for COVID-19 diagnosis using NP, nasal, and buccal and nasal swab samples were selected to investigate the analytical and clinical efficacy of these clinical samples.

The majority of RTK validation studies were conducted prior to the emergence of SARS-CoV-2 variants of concern (VOC) [7,16,27]. There is a lack of data on VOCs’ routine diagnostic performance to date. Furthermore, clinical validation studies comparing multiple VOCs in parallel are hardly feasible [27]. The dominant VOC during the current study’s duration was the Delta variant. Most of the previous studies investigated the clinical efficacy of RTK devices [16,19,20], but our assessment was on NP, nasal, and buccal and nasal swab samples from suspected COVID-19 patients in a public sector hospital setting. A previous study on the validation of RTKs using buccal and saliva samples for COVID-19 diagnosis, performed by Ku et al. (2021), reported a very high PPV (100%) with a comparatively low NPV (48%) for buccal swabs compared to NP swabs [28]. However, in the current study, 98% PPV and 100% NPV were reported for buccal and nasal swab samples compared to NP swabs. In another study, the authors reported the poor sensitivity of buccal swabs for COVID-19 diagnosis among children compared to RT-PCR [29]. 

Different kits used for the qRT-PCT may affect the outcome as well. In the current study, three cases were positive using RTKs, but qRT-PCR results showed no detection for all targeted genes. After repeat extraction, the result showed a high Ct value detected for one target gene. When a different RT-PCR kit was used, the same sample gave a positive result on all target genes. This could be due to the low viral load and the quality of RNA samples that different kits can detect. Apart from this, RTKs and qRT-PCR detect different molecules. RTK-Ag detects SARS-CoV-2 viral proteins to determine active infection, while qRT-PCR identifies genetic material (RNA) from the virus, even when the virus is shedding and inactive. Hence, the current study suggests that it is not appropriate to use RT-PCR as a gold-standard method to correlate the results with the RTKs.

Although RTKs cannot replace the qRT-PCR as a gold-standard diagnostic test to confirm or rule out the presence of SARS-CoV-2, our analytical and clinical data indicate that they may be employed in smaller clinical and household settings. A recent study conducted by Patriquin et al. (2022) showed 88% sensitivity of nasal swabs compared with RTKs using NP swabs [19]. In another study, RTKs using nasal swabs were comparable to qRT-PCR using NP swabs [12]. Kojima et al. (2019) reported an 85% detection rate for nasal and 79% for NP swabs [26]. However, the current study showed 96% sensitivity for nasal and 98% for NP and for nasal and buccal swabs compared with RT-PCR using NP swabs. Hence, our study validated the efficacy of nasal, and buccal and nasal samples in the community for easy sample collection and early diagnosis. 

To our knowledge, no study has been published to evaluate the efficacy of buccal and nasal swabs together for the early diagnosis of COVID-19. Ku et al. (2021) conducted a study to evaluate the efficacy of buccal and saliva samples together, showing moderate agreement compared with NP swabs [28]. In the present study, the nasal and buccal swabs showed 98% accuracy compared with the results of qRT-PCR using NP swabs. Hence, the current study recommends that the initial screening for COVID-19 can be carried out by using buccal and nasal samples together.

Because of the high sensitivity of RT-PCR, doubts have been raised about the clinical and epidemiological implications of being positive for COVID-19 infection [23] since a low quantity of viral RNA may still be detected by RT-PCR months after infections in certain cases, indicating that the transmission potential is questionable [7]. The recent literature reports that the expected duration for COVID-19 infection may start after four days of exposure and remain up to 10 to 14 days [30]. Because RTKs consistently identify patients with high viral loads, they might be beneficial in screening efforts to detect the antigen in COVID-19-infected individuals [8,21].

The WHO recommends using RTKs to support case and contact diagnosis during large outbreaks. They can also monitor disease incidence trends, especially in remote areas and areas with large populations, such as schools, childcare facilities, and prisons [6]. Many studies have reported the efficacy of NP swabs for the detection of antigens using RTKs [21,29,31], but the current study’s findings suggest that RTK devices using nasal and buccal samples compared to NP swabs may provide similar results with an easier approach for diagnosing COVID-19. Furthermore, the nasal and buccal and nasal swabs are easier to collect compared to NP swabs, allowing researchers to assess the feasibility of widespread community application by evaluating the patient acceptability for different types of samples; we found that the collection of buccal and nasal swabs was more manageable compared to NP swabs. Apart from the advantages of the current study, there were also some limitations, as we conducted this study on a comparatively smaller number of individuals and in one hospital only.

## 5. Conclusions

In conclusion, the RTK tests using nasal and buccal, and nasal swabs were comparable with qRT-PCR for the early diagnosis of COVID-19. Our results showed that the primary screening of COVID-19 can be done using nasal and buccal, and nasal swab-based RTKs. The present study supports the use of nasal and buccal, and nasal swabs as an alternative medium to NP swabs for accurate clinical diagnosis since these are more cost-effective and easier to collect. Overall, the nasal and buccal and nasal-based RTKs can be utilized for early and rapid diagnosis to lower the risk of community spread and transmission.

## Figures and Tables

**Figure 1 diagnostics-12-00847-f001:**
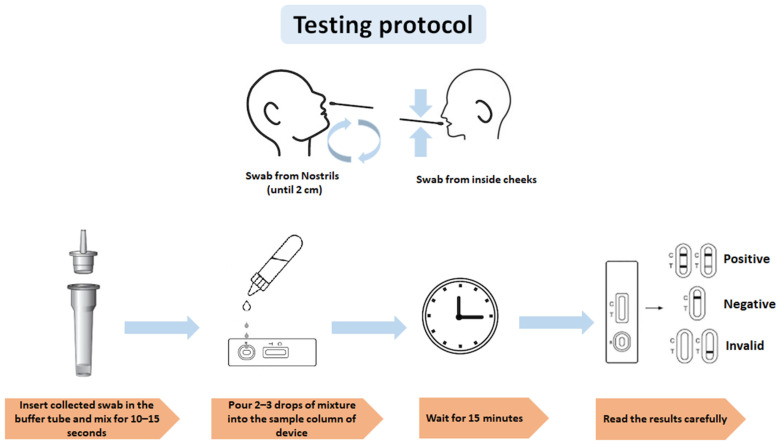
Testing protocol for RTK test using nasal and buccal swabs.

**Table 1 diagnostics-12-00847-t001:** Different types of clinical specimens used in current study.

Specimen Type	RTK Test	RT-PCR (With Ct Value)
Nasopharyngeal swab	157	157
Buccal and nasal swab	57	-
Nasal swab	100	-

**Table 2 diagnostics-12-00847-t002:** Demographic characteristics of study participants.

Characteristics		qRT-PCR (*n*-157)	
		Detected	Not Detected
Gender	Male	52	38
	Female	50	17
Age (Years)	9–15	5	0
	16–30	29	3
	31–45	25	8
	46–60	19	11
	>61	24	33
Ethnicity	Malay	89	51
	Chinese	9	4
	Indian	4	0

**Table 3 diagnostics-12-00847-t003:** Sensitivity, specificity, accuracy, PPV, and NPV of different respiratory samples in comparison with qRT-PCR.

Test Evaluation	Nasal and Buccal	Nasal	Nasopharyngeal
Sensitivity	98%	96%	98%
Specificity	100%	98%	94%
Accuracy	98%	97%	97%
Positive Predictive Value (PPV)	100%	98%	97%
Negative Predictive Value (NPV)	75%	96%	96%

**Table 4 diagnostics-12-00847-t004:** Association of different tested samples in comparison to RT-PCR as a reference method.

Type of Samples		qRT-PCR		Total	Fisher’s Exact Test (*p*-Value)
		Detected	Not Detected		
Nasal and Buccal	Positive	53	0	53	<0.001
	Negative	1	3	3	
	Total	54	3	57	
Nasal	Positive	49	1	50	<0.001
	Negative	2	48	50	
	Total	51	49	100	
Nasopharyngeal	Positive	101	2	103	<0.001
	Negative	3	51	54	
	Total	104	53	157	

**Table 5 diagnostics-12-00847-t005:** Comparison of different tested samples with qRT-PCR Ct values.

RT-PCR (Ct Value)	Number of Samples (Positive with qRT-PCR) (*n*-104)	Nasal and Buccal Swabs (*n*-53)	Nasal Swabs (*n*-50)	Nasopharyngeal Swabs (*n*-103)
<20.00	33 (31.73%)	8 (7.69%)	25 (24.03%)	32 (30.76%)
20.1–23.5	16 (15.38%)	8 (7.69%)	8 (7.69%)	16 (15.38%)
23.6–26.0	9 (8.65%)	5 (4.80%)	4 (3.84%)	9 (8.65%)
26.1–29.5	17 (16.34%)	11 (10.57%)	6 (5.76%)	17 (16.34%)
29.6–32.0	11 (10.57%)	8 (7.69%)	2 (1.92%)	11 (10.57%)
32.1–35.5	7 (6.73%)	6 (5.76%)	5 (4.80%)	7 (6.73%)
35.6–38.0	7 (6.73%)	4 (3.84%)	0	4 (3.84%)
38.1–41.5	4 (3.84%)	3 (2.88%)	0	2 (1.92%)

## Data Availability

Data related to the study can be shared upon reasonable request to the corresponding author.

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
