# Peer review of "A Performance Assessment Study of Different Clinical Samples for Rapid COVID-19 Antigen Diagnosis Tests"

_diagnostics, 2022, doi:10.3390/diagnostics12040847_

Round 1
Reviewer 1 Report
Please refer to the attachments.

Author Response
Reviewer 1
Comments and Suggestions for Authors
Title: The authors need to revise the title of the paper in a more meaningful way.
Response: (Page 1, Line 3) The title has been revised as “A performance assessment study of different clinical samples for rapid COVID-19 antigen diagnosis tests”.
Abstract: The abstract is written in a way lacks logic. It should highlight the salient findings more critically. Based on what was presented by the authors, I suggest inserting an introduction sentence and a conclusion sentence in the abstract.
Response: (Page 1, Line 16-32) The abstract has been revised and sentences about introduction and conclusion of the study has been added.
Keywords are present in the title (COVID-19), choose others.
Response: (Page 1, Line 33) The keyword ‘COVID-19’ is deleted.
Introduction need more convincing rational for this article.
Response: The study rationale has been added in the last paragraph of introduction section (Page 2, Line 46-47) “Depending on the type of methods or kits used, the qRT-PCR takes 1-4 hours to complete. In addition, it must be carried out by trained molecular personnel ”
(Page 2, Line 76-81) “Currently, the large gap between the high number of cases and the qRT-PCR testing capacity is a major limitation, resulting in a need for alternative assays such as RTKs, which may identify the presence of the antigen in different respiratory samples. Many types of RTKs have been recently developed and commercially available. However, the information about the performance efficacy of these assays with different clinical sam-ples is limited. Keeping in mind the feasibility of RTKs with different clinical samples, we conducted the current study to evaluate the accuracy of different respiratory sam-ples used for COVID-19 early diagnosis by various RTKs. The efficacy of different clin-ical samples (with RTK test) was compared with the qRT-PCR.”
Methods: Provide statistical experimental work plan at the start of M&M. No detail description is available about the experimental design.
Response: (Page 4, Line 148-153) Our statistical analysis was stated at 2.6 Statistical analysis.
Results: The results of this study are not fully explained therefore the interpretation of the results is very difficult. The author needs to provide the % increase or decrease rather than just writing ''significantly increased….''.
Response: (Page 5, Line 160, 164-166, 172-173 and 176-180) The result section has been revised thoroughly. Furthermore, the values and percentages has been written wherever applicable.
In tables 2, 3, 4 and 5 Standard error or standard deviation? Please specify and improve tables captions.
Response: The descriptive statistics can not be performed as the data variable are “Positive” or “Negative”. For the ages of study participants, we categorized the data in age groups, that’s why the std. deviation cannot be mentioned. Furthermore, the table captions are improved wherever applicable.
Discussion: Authors should discuss the results integrally. The discussion is based on individual results. I suggest that integrating the results will give more value to the work.
Response: (Page 6, Line 187-196, 206-232 and 243-253) The discussion section has been revised thoroughly.
The discussion is poorly written hence, needs rewriting. The discussion should be further strengthened by adding some more relevant papers. The literature search is insufficient, only few related research papers in the past three years are cited, add the latest research results appropriately. See the below links if you think it will benefit your discussion.
Response: Dear Reviewer we didn’t found any link in your report. Might be a technical error while uploading the comments. Furthermore, we have revised the discussion section and added more relevant studies from the recent years in this section.
Conclusion: Rewrite the conclusion. It needs to be much improved.
Response: (Page 7, Line 257-261)

Reviewer 2 Report
Thank you for the opportunity to review your manuscript.
The article's title references bridging the gap between diagnosis and isolation. This does not appear to be the goal of the article. Please consider a more accurate title.
In the Introduction -
- It is stated SARS-CoV-2 specific antigen testing "was just recently established" - the first commercial assays received regulatory approval in mid-2020.
- Covid-19 is stated to be an epidemic, not pandemic.
- The article reports "a scarcity of scientific literature reporting their accuracy" regarding antigen detection kits - a systematic analysis and meta-analysis published in mid-2021 identified 29 relevant studies
The study undertaken is described as a cross sectional evaluation of suspected COVID-19 cases in section 2.1, but as a retrospective assessment in section 4. Please clarify.
Materials and Methods
"Suspected cases" is not defined.
When nasal and buccal swabs were collected, were they placed in the same tube/buffer?
Data regarding day of illness when samples were collected, or VOC dominant during the time of this study are not presented.
Discussion
The phrase, "theories about COVID-19 are still unknown" needs to be reworked.
This study only evaluated persons suspected to have COVID-19; therefore, does not address accuracy or utility in using RDKs in asymptomatic patients.
The Discussion and Conclusions sections speaks to patient acceptibility for different samples, patient preference, comfort and convenience; however, data is not presented to suggest that information was surveyed or analyzed.
Author Response
Reviewer 2
Comments and Suggestions for Authors
Thank you for the opportunity to review your manuscript.
Response:
The article's title references bridging the gap between diagnosis and isolation. This does not appear to be the goal of the article. Please consider a more accurate title.
Response: (Page 1, Line 2-3) The title has been changed in the revised version of manuscript.
In the Introduction -
- It is stated SARS-CoV-2 specific antigen testing "was just recently established" - the first commercial assays received regulatory approval in mid-2020.
Response: (Page 2, Line 64-65) The sentence has been rephrased with a reference.
- Covid-19 is stated to be an epidemic, not pandemic.
Response: revised through-out the manuscript
- The article reports "a scarcity of scientific literature reporting their accuracy" regarding antigen detection kits - a systematic analysis and meta-analysis published in mid-2021 identified 29 relevant studies
Response: (Page 2, Line 67-68) The sentence has been edited in the revised version of manuscript. “Unfortunately, there is a lack of scientific literature supporting their accuracy, and real-time performance using different types of clinical samples.”
The study undertaken is described as a cross sectional evaluation of suspected COVID-19 cases in section 2.1, but as a retrospective assessment in section 4. Please clarify.
Response: (Page 6, Line 194) The sentence has been corrected.
Materials and Methods
"Suspected cases" is not defined.
Response: (Page 2, Line 89-90). A sentence has been added to define the suspected cases.
When nasal and buccal swabs were collected, were they placed in the same tube/buffer?
Response: Yes, when nasal and buccal swabs were collected together, these were placed in the same buffer. At first the nasal swab was collected and then mixed in the buffer, then with a new swab, a buccal swab was collected and mixed in the same buffer.
Data regarding day of illness when samples were collected, or VOC dominant during the time of this study are not presented.
Response: (Page: 7, Line: 192-195) Data regarding the day of illness was one of the study limitations. It was not available. Furthermore, during the study duration, In Malaysia we have the Delta strain. The first case of Omicron was reported in Last week of December. A paragraph about VOC /s RTKs has been added in the revised version of manuscript.
Discussion
The phrase, "theories about COVID-19 are still unknown" needs to be reworked.
Response: (Page 7, Line 236) The statement has been removed from the revised version of manuscript.
This study only evaluated persons suspected to have COVID-19; therefore, does not address accuracy or utility in using RDKs in asymptomatic patients.
Response: The discussion statements about asymptomatic patients has been removed.
The Discussion and Conclusions sections speaks to patient acceptibility for different samples, patient preference, comfort and convenience; however, data is not presented to suggest that information was surveyed or analyzed.
Response: (Page 7 and 8, Line 187-196, 206-232, 239, 243-253, and 257-261) The discussion section has been thoroughly revised and all of the irrelevant information has been removed from the revised version of manuscript. Furthermore, the statements about the patient satisfaction have been removed from the conclusion section.
Reviewer 3 Report
The manuscript entitled “A performance assessment study of different clinical samples for rapid COVID-19 diagnosis tests: Bridging the gap between diagnosis and isolation” by Ahmed et al was submitted to diagnostics for consideration as a regular article. The authors describe the results of COVID testing from a cohort of 157 patients from a public hospital population in Malaysia using PCR and antigen based methods. The general claim is that various specimen types can be used but some perform better than others. While interesting, this manuscript lacks sufficient detail to determine the validity of this claim. Many manuscripts have been published on similar topics, so providing these details is necessary to elevate this paper. My comments are summarized below:
- The title does not describe the work performed. “Bridging the gap between diagnosis and isolation: would require quantification of the gap and a demonstration of improvement. It is not clear if this paper does that. “antigen” is missing after “rapid” since the authors did not evaluate rapid PCR tests
- Why are barber shops specifically mentioned in the abstract? This seems out of place. What about schools, places of work, the immunocompromised, pre-op patients?
- Introduction: 2nd paragraph, first line. Many tests on the market take less than 4 hours. The Cepheid test mentioned by the authors takes roughly 1 hour from sample to answer.
- Introduction: 4th paragraph. SARS-CoV-2 is a “pandemic” not an “epidemic”
- Methods 2.1. There is no exclusion for age group. Infant and toddler anatomy may be sufficiently unique and may result in inaccurate results. What is the rationale for including?
- Table 1: What is nasal & buccal and how is this different than nasal alone? Same swab for both locations or same VTM for both swabs? The table can just be stated in the text.
- Methods 2.2. If all NP swabs went to PanBio and everything else went to ProDetect, how can you compare NP to other specimens. Two different tests may have entirely different sensitivities.
- Methods 2.3. Which RTK is described? Subtitle “testing protocol” has no real meaning to the reader. “RTK Method” may be better.
- Figure 1 says “self-conduct”. Were these self-collected, provider observed, or provider performed?
- Methods 2.5. Which specimens went to which PCR assay? (also same comment for Table 2). This is critical if you are comparing against performance. The GeneXpert test does not contain an internal control to rule out false negatives
- Methods 2.6. NPV and PPV are not calculated in any table in the paper.
- Following table 4, the statement claims “no statistical difference was found…” Where is the data for this?
- Table 5 compares CT values for RDRP. What about tests that give 2 CT values or do not contain RDRP. It is unclear what is being shown here.
- Patients showed preference to nasal and buccal? This is a “result” and should be supported with evidence. The last sentence is completely unsubstantiated based on the data presented.
Author Response
Reviewer 3
Comments and Suggestions for Authors
The manuscript entitled “A performance assessment study of different clinical samples for rapid COVID-19 diagnosis tests: Bridging the gap between diagnosis and isolation” by Ahmed et al was submitted to diagnostics for consideration as a regular article. The authors describe the results of COVID testing from a cohort of 157 patients from a public hospital population in Malaysia using PCR and antigen-based methods. The general claim is that various specimen types can be used but some perform better than others. While interesting, this manuscript lacks sufficient detail to determine the validity of this claim. Many manuscripts have been published on similar topics, so providing these details is necessary to elevate this paper. My comments are summarized below:
Response: Thank you for your valuable suggestions and comments which makes the manuscript better for the readers. We have addressed all of the comments provided. Furthermore, the manuscript has been checked thoroughly for English proofreading.
The title does not describe the work performed. “Bridging the gap between diagnosis and isolation: would require quantification of the gap and a demonstration of improvement. It is not clear if this paper does that. “antigen” is missing after “rapid” since the authors did not evaluate rapid PCR tests
Response: (Page 1, Line 2-3) The title has been revised as “A performance assessment study of different clinical samples for rapid COVID-19 antigen diagnosis tests”.
Why are barber shops specifically mentioned in the abstract? This seems out of place. What about schools, places of work, the immunocompromised, pre-op patients?
Response: (Page 1, Line 27-32) The abstract has been revised in the manuscript.
Introduction: 2nd paragraph, first line. Many tests on the market take less than 4 hours. The Cepheid test mentioned by the authors takes roughly 1 hour from sample to answer.
Response: (Page 2, Line 46-47) The sentence has been rephrased as “Depending on the type of methods or kits used, the qRT-PCR takes 1-4 hours to complete. In addition, it must be carried out by trained molecular personnel”.
Introduction: 4th paragraph. SARS-CoV-2 is a “pandemic” not an “epidemic”
Response: Revised through-out the manuscript
Methods 2.1. There is no exclusion for age group. Infant and toddler anatomy may be sufficiently unique and may result in inaccurate results. What is the rationale for including?
Response: (Page 5, Table 5) The rationale behind including all age group patients was to see the efficacy in every age group. As in most of the hospital settings in Malaysia, the COVID-Ag or PCR is mandatory for all of the age groups without any exclusion. Furthermore, in the current study we didn’t find any infant (less than 1 year of age) and toddler (less than 3 year of age) during the study duration.
For the convenience of readers, the age group 1-15 has been changed as “9-15” as in this age group the participant ages were from 9 year to 15 year. The remaining age groups have not been changed.
Table 1: What is nasal & buccal and how is this different than nasal alone? Same swab for both locations or same VTM for both swabs? The table can just be stated in the text.
Response: In the current study, we evaluated different kits using different type of clinical samples. The buccal and nasal means two different swabs mixed in one buffer and then the test was performed. While in nasal alone, only the nasal swab sample was mixed with the buffer and tested. These (nasal & buccal and nasal) swabs didn’t contain VTM, as the test was performed immediately after sample collection. The swabs containing VTM were used only for RT-PCR using NP swabs.
Methods 2.2. If all NP swabs went to PanBio and everything else went to ProDetect, how can you compare NP to other specimens. Two different tests may have entirely different sensitivities.
Response: We agree that two different kits may have entirely different sensitivities, but as we know that the PanBio devices are approved from WHO and they use only Nasopharyngeal swab for the testing but in ProDetect we can use nasal and saliva both. Hence, the aim of current study was to evaluate the efficacy of samples used by different kits including Panbio and ProDetect rather than to evaluate the performance efficacy of the kits. Previously there are many studies conducted to see the efficacy of these devices.
Methods 2.3. Which RTK is described? Subtitle “testing protocol” has no real meaning to the reader. “RTK Method” may be better.
Response: (Page 3, Line 124) The subheading has been changed as “RTK method”. The described RTK method is of ProDetect, which is also quite same for Panbio.
Figure 1 says “self-conduct”. Were these self-collected, provider observed, or provider performed?
Response: (Page 4, Line 133) The legend has been rewritten as “Figure 1. Testing protocol for RTK test using buccal and nasal swabs”.
Methods 2.5. Which specimens went to which PCR assay? (also same comment for Table 2). This is critical if you are comparing against performance. The GeneXpert test does not contain an internal control to rule out false negatives
The GeneXpert test have two internal control; A Sample Processing Control (SPC) and a Probe Check Control (PCC) are also included in the cartridge utilized by the GeneXpert instrument. The SPC is present to control for adequate processing of the sample and to monitor for the presence of potential inhibitor(s) in the RT-PCR reaction. The SPC also ensures that the RT-PCR reaction conditions (temperature and time) are appropriate for the amplification reaction and that the RT-PCR reagents are functional. The PCC verifies reagent rehydration, PCR tube filling, and confirms that all reaction components are present in the cartridge including monitoring for probe integrity and dye stability. As each individual test is performed, the Instrument Status, Sample Processing, Integrity of PCR Reagents, and PCR Efficiency are evaluated. These internal quality control features verify multiple aspects of assay performance for every sample tested”. –By Xpert® Xpress SARS-CoV-2, Cepheid
Methods 2.6. NPV and PPV are not calculated in any table in the paper.
Response: (Page 5, Table 3) PPV and NPV have been calculated and written in table 3.
Following table 4, the statement claims “no statistical difference was found…” Where is the data for this?
Response: (Page 6, Line 176-180) The statement “no statistical difference was found…” is for Table 5, but not for table 4. Furthermore, the following sentence has been added in the beginning of paragraph:
“The overall Ct values obtained from qRT-PCR have been shown in table 5, which also shows the comparison of Ct values with the positive ratio of different clinical samples.”
Table 5 compares CT values for RDRP. What about tests that give 2 CT values or do not contain RDRP. It is unclear what is being shown here.
Response: (Page 6, Line 182) The “RDRP” has been removed from the legend as it was a typing mistake before. The shown Ct values are the overall Ct values obtained from qRT-PCR, which also shows the comparison of Ct values with the positive ratio of different clinical samples.
Patients showed preference to nasal and buccal? This is a “result” and should be supported with evidence. The last sentence is completely unsubstantiated based on the data presented.
Response: (Page7, Line 258-261) The sentence “When comparing nasal and buccal swabs to NP swabs in terms of comfort and convenience, patients showed preference to nasal and buccal swabs” has been removed from the conclusion section as it was an observation during the sample collection but not supported by any of the result in the current study.
Reviewer 4 Report
the manuscript covers an original and interesting topic about the sensitivity and specificity of COVID-19 tests.
the manuscript is well written, the methods clearly presented and results well organized.
discussion and conclusions are coherent with results
Author Response
Reviewer 4
Comments and Suggestions for Authors
the manuscript covers an original and interesting topic about the sensitivity and specificity of COVID-19 tests.
the manuscript is well written, the methods clearly presented and results well organized.
discussion and conclusions are coherent with results
Response: We want to say thanks to the reviewer for his/her valuable time to review our manuscript and give his/her comments. We hope that this manuscript will be good for the readers and community to choose the best and accurate RTK, based on different sample types.
Round 2
Reviewer 3 Report
The authors have addressed my concerns